# From SARS-CoV-2 to Global Preparedness: A Graphical Interface for Standardised High-Throughput Bioinformatics Analysis in Pandemic Scenarios and Surveillance of Drug Resistance

**DOI:** 10.3390/ijms25126645

**Published:** 2024-06-17

**Authors:** Tomas Cumlin, Ida Karlsson, Jonathan Haars, Maria Rosengren, Johan Lennerstrand, Maryna Pimushyna, Lars Feuk, Claes Ladenvall, Rene Kaden

**Affiliations:** 1Department of Medical Sciences, Section for Clinical Microbiology, Uppsala University, Akademiska Sjukhuset Entrance 40, 751 85 Uppsala, Sweden; 2Clinical Genomics Uppsala, Science for Life Laboratory, Uppsala University, 751 85 Uppsala, Sweden; 3National Genomics Infrastructure Uppsala, Uppsala University, 751 08 Uppsala, Sweden; 4Department of Immunology, Genetics and Pathology, Uppsala University, 751 08 Uppsala, Sweden

**Keywords:** automated bioinformatics, pandemic preparedness, high-throughput sequence data analysis, outbreak surveillance, antimicrobial resistance surveillance

## Abstract

The COVID-19 pandemic highlighted the need for a rapid, convenient, and scalable diagnostic method for detecting a novel pathogen amidst a global pandemic. While command-line interface tools offer automation for SARS-CoV-2 Oxford Nanopore Technology sequencing data analysis, they are inapplicable to users with limited programming skills. A solution is to establish such automated workflows within a graphical user interface software. We developed two workflows in the software Geneious Prime 2022.1.1, adapted for data obtained from the Midnight and Artic’s nCoV-2019 sequencing protocols. Both workflows perform trimming, read mapping, consensus generation, and annotation on SARS-CoV-2 Nanopore sequencing data. Additionally, one workflow includes phylogenetic assignment using the bioinformatic tools pangolin and Nextclade as plugins. The basic workflow was validated in 2020, adhering to the requirements of the European Centre for Disease Prevention and Control for SARS-CoV-2 sequencing and analysis. The enhanced workflow, providing phylogenetic assignment, underwent validation at Uppsala University Hospital by analysing 96 clinical samples. It provided accurate diagnoses matching the original results of the basic workflow while also reducing manual clicks and analysis time. These bioinformatic workflows streamline SARS-CoV-2 Nanopore data analysis in Geneious Prime, saving time and manual work for operators lacking programming knowledge.

## 1. Introduction

The COVID-19 pandemic revealed the necessity for a rapid, convenient, and scalable establishment of diagnostic methods for a novel pathogen during a worldwide pandemic. Between 24 January 2020 and 24 January 2021, the number of confirmed coronavirus cases worldwide surged from 1734 to 100,138,169 [1], significantly increasing the workload on healthcare personnel, for example, in patient diagnosis [2]. In response, one favourable approach employed was real-time reverse transcription–polymerase chain reaction (RT-PCR) for fast and accurate diagnosis of SARS-CoV-2 nucleic acids in nasopharyngeal fluids [3]. Furthermore, obtaining the virus’s genetic sequence can contribute to genetic variation tracking and enhance understanding of novel mutations and managing their implications [4]. With a demand to detect single nucleotide variants (SNVs), while also considering pragmatic factors like turnaround time, cost, and sensitivity, long-read sequencing using the Oxford Nanopore Technology platform (ONT) emerged as a viable technology to sequence regions of interest in SARS-CoV-2 [5,6,7]. At this stage, ONT sequence data quality has reached a level adequate for SNV detection in SARS-CoV-2 genomes [8]. To target the SARS-CoV-2 genomes in a patient sample for sequencing, multiplex PCR tiling methods can be used.

A wide range of bioinformatic tools is available for the computational analysis of sequence data, performing common practices such as read assembly, quality control checks, variant calling, and phylogenetic assignment. Many of these tools are designed to operate within a command-line interface (CLI), enabling easy integration into pipelines to automate bioinformatic analyses. This automation reduces manual tasks and the risk of human error, while enabling rapid processing of large volumes of patient samples in clinical practices. Examples of such pipelines include the nCoV-2019 novel coronavirus bioinformatics protocol [9] from the Artic Network, NanoCoV19 [10], poreCov [11], and Viralrecon [12], which perform raw read preprocessing, alignment, variant calling, and generate consensus sequences from SARS-CoV-2 ONT sequence data. However, the practicality provided by these pipelines remains inaccessible to users with limited programming skills.

Bioinformatic software with a graphical user interface (GUI) enables laboratories to conduct sequencing experiments, including routine bioinformatics analysis, requiring only a small investment in training the laboratory staff. An example of such software is Geneious Prime (www.geneious.com, accessed on 17 October 2023), which is available as a GUI platform. It features functions such as sequencing read assembly, coverage calculations of sequence alignments, and SNV detection. Additional tools can be imported as plugins and utilised in the software. Geneious Prime also offers the option to compile its functions and software into automated workflows, thereby enabling the automation of bioinformatic analyses within a GUI environment.

The purpose of this work was to provide fast and reliable workflows that automate the bioinformatics analysis of SARS-CoV-2 ONT sequencing data, requiring no programming knowledge to operate. With that, two workflows were developed to run in Geneious Prime 2022.1.1 [13]. With regard to potential new pandemics, these workflows can be easily modified for the analysis of other microorganisms or test data obtained from different sequencing platforms. Our objective has been to minimise setup configurations and ensure that the workflows are compatible across multiple operating systems (Windows, Linux, and macOS). Since 2023, both workflows have been integrated into routine use within the Clinical Microbiology unit at Uppsala University Hospital. They are used for SARS-CoV-2 analysis of ONT sequenced patient samples and for national surveillance by the Public Health Agency of Sweden.

## 2. Results

Two workflows were created in Geneious Prime’s GUI to standardise and streamline the bioinformatics analysis of SARS-CoV-2 sequencing data for users with limited programming skills. The basic workflow, named ‘SARS-CoV-2_Assembly_Basic’, performs trimming, alignment, generation of a consensus sequence, and annotations. The second workflow, named ‘SARS-CoV-2_Assembly_WrapperPlugins’, includes two additional steps that provide phylogenetic assignments for the test data. For these steps, two plugins were developed for this workflow using pangolin (version 4.0) [14] and Nextclade CLI (version 2.14.0) [15]. After installation and importation of FASTQ files, the workflow allows the user to perform automatic bioinformatics analysis of 96 samples (or more) with one single mouse click.

### 2.1. Workflow Validation

The workflows and plugins successfully conducted the bioinformatics analysis of SARS-CoV-2 ONT sequencing data in Geneious Prime 2022.1.1 on Linux (Kernel version: 5.15.0-58-generic), Mac (macOS Monterey 12.6.3), and Windows 10 Enterprise (Version 19044.2251). The original workflow with the basic analysis steps was previously validated through comparison with the method used by the Public Health Agency of Sweden by analysing 30 sequences in parallel. As validation criteria, we determined that no discrepancies are allowed. We continued following our results with the pango-lineage determination that was automatically executed for our files after uploading the sequences to GISAID.

From the input FASTQ files, the workflow saved the alignments and consensus sequences, with annotations added, within Geneious (www.geneious.com, accessed on 17 October 2023). The consensus sequences were exported to a selected desktop folder as FASTA files.

The wrapper plugins created in Geneious Prime, using pangolin and Nextclade CLI, underwent testing on Linux, macOS, and Windows using the whole genome of SARS-CoV-2 Wuhan-Hu-1. By using Docker via a Python script, one pangolin plugin could be used across all operating systems. While the Python script of the Nextclade plugin functioned on both Linux and macOS, a Batch script had to be included for the Nextclade plugin to execute the Python script on Windows. Therefore, there are two wrapper plugins for Nextclade: one for Linux and macOS, and one for Windows. Once finalised, the wrapper plugins could run in Geneious on the three operating systems and accurately assign the pango-lineage B. Subsequently, the wrapper plugins were integrated into the basic workflow, thus creating the advanced workflow, ‘SARS-CoV-2_Assembly_WrapperPlugins’.

The advanced workflow with plugins was validated using 96 clinical samples, all resulting in accurate diagnoses that matched the results obtained from the validated basic workflow, which had utilised the web interfaces of pangolin and Nextclade (Table 1). An experienced biomedical analyst successfully analysed 96 samples using the ‘SARS-CoV-2_Assembly_Basic’ workflow in 63 min. The process of obtaining the pango-lineage assignments using the web interfaces of pangolin and Nextclade required an additional 6 min. Thus, the total analysis time amounted to 69 min. Running the ‘SARS-CoV-2_Assembly_WrapperPlugins’ workflow in Geneious Prime took 56 min, suggesting a slight reduction in analysis time with the implementation of the plugins.

### 2.2. Test Data

The test data consisting of 18 ONT sequenced SARS-CoV-2 samples, used for workflow validation, were made available at the Sequence Read Archive (SRA) at NCBI (Accession number: PRJNA1048178). The removal of all raw reads not aligning with the SARS-CoV-2 reference sequence successfully eliminated human DNA from the samples, securing the privacy rights of the patients. The percentage of mapped reads, among the filtered samples, against the SARS-CoV-2 reference sequence (NC_045512), ranged from 93.1% to 98.91%, with a group average of 97.12%. Two of the filtered samples exhibited no reads mapped against the human reference genome assembly GRCh38, while the other 16 samples demonstrated a range of 0.09% to 0.47% mapped reads and a group average of 0.3%. Running the filtered samples with the workflow ‘SARS-CoV-2_Assembly_WrapperPlugins’ gave the same pango-lineages from pangolin and Nextclade as the original, unfiltered samples, confirming that no important reads had been discarded during the filtering process.

### 2.3. Visual Inspections in Geneious Prime

As a GUI software (www.geneious.com, accessed on 17 October 2023), Geneious Prime facilitates visual inspection of the output generated by the workflows. The read pile-up in the coverage track allows assessment of sequencing coverage across various regions for each sample, offering a valuable tool for identifying weaknesses or potential issues in the sequencing protocol, such as faulty or missing primers (Figure 1).

Annotations for specific mutations are displayed in the consensus sequences at their respective positions in the genome (Figure 2). This enables confirmation of pango-lineage classifications, detection of new mutations, identification of recombination cases, or recognition of contamination events. Confirming the presence of a specific mutation in a sequence is useful when certain mutations are considered more important than the lineage itself. This is especially relevant in cases where a mutation confers drug resistance or is the subject of specific monitoring. Annotations can be searched for in Geneious Prime or filtered by name or type. This allows for rapid monitoring of mutations conferring particular traits such as drug resistance or as markers during epidemiological investigations and contact tracing.

## 3. Discussion

The COVID-19 pandemic has emphasised the necessity for automating the analysis of SARS-CoV-2 genome sequencing data, due to the high volume of patient samples. Efforts to streamline such processes, including bioinformatics analysis steps, have yielded various workflows, such as Artic Network’s nCoV-2019 novel coronavirus bioinformatics protocol, NanoCoV19, poreCov, and Viralrecon. However, like many bioinformatics pipelines, these workflows run in a CLI and require programming knowledge. To address this limitation, the workflows we present are designed specifically for the Geneious Prime software’s GUI (www.geneious.com, accessed on 17 October 2023). This enables users with limited programming skills to analyse SARS-CoV-2 sequencing data, while preserving the benefit of automated analysis of multiple samples with few manual steps. Another advantage is the convenience of consolidating data storage, analysis, and results visualisation within a single software platform. One limitation to acknowledge is the workflows’ dependency, and thus constrictions, on the current capacity of Geneious Prime. For example, there is no present solution to import the workflow without the need to link data files to associated analysis steps and to manually implement the plugins into the workflow that performs phylogenetic assignment.

Experienced personnel at Uppsala University Hospital confirmed that running the workflow with the wrapper plugins generated compatible results compared to manually running the software pangolin (cov-lineages.org, accessed on 16 June 2024) and Nextclade (nextstrain.org, accessed on 21 December 2023) in their respective web interfaces. Personnel responsible for running the SARS-CoV-2 Nanopore workflow described the workflow with plugins as more efficient and user-friendly due to the reduction in manual steps required for the final pango-lineage assignment. Using pangolin and Nextclade as plugins also eliminates potential issues, such as limited internet access, slow access to pangolin’s web interface due to high traffic load, and the risk of human errors occurring in manual steps. These steps involve transferring the generated FASTA files into the web tools and later copying the results back to a results sheet. Automating these steps ensures that the phylogenetic assignment and documentation are performed according to the validated routine with higher confidence, which is a crucial aspect in clinical diagnostics. A high number of manual steps can also be wearing on the staff, time-consuming for less experienced personnel, and require longer training time. Another advantage is documentation, as the plugins will save the versions of pangolin and Nextclade CLI in the result documents generated from the workflow. Finally, automation allows for better tracking of previous jobs, as the setup of the analysis steps remains constant.

The workflows were utilised in the laboratory of Clinical Microbiology and Hospital Hygiene at Uppsala University Hospital for the bioinformatics analysis of SARS-CoV-2 tests in Uppsala County. Between April 2021 and December 2023, the basic workflow was applied to 12,483 sequenced SARS-CoV-2 patient samples for national surveillance. The ability to track individual mutations through annotations was used for contact tracing in a study which linked several cases on the same flight by the S:E484Q mutation [16]. The workflows have also been used for research projects which investigated circulating variants of concern and mutations causing resistance against monoclonal antibodies used in the treatment of COVID-19 [17,18].

The workflows were created as a tool for global preparedness. Since the basic functions of the workflows involve trimming, aligning, and annotating amplicon sequencing data, they are not limited to the analysis of SARS-CoV-2 but can be customised for other organisms. By altering the primers, reference genome, and annotation features, it is possible to execute the same workflow for other pathogens, enabling quick access to an analysis workflow in future outbreaks of a novel pathogen.

## 4. Materials and Methods

### 4.1. Geneious Workflows

Two workflows were developed using Geneious Prime 2022.1.1, named ‘SARS-CoV-2_Assembly_Basic’ and ‘SARS-CoV-2_Assembly_WrapperPlugins’. Each performs the same basic bioinformatics analysis, with the latter one also providing phylogenetic assignment of the samples using the pangolin and Nextclade tools as plugins (see Section 4.1.1. Workflow structure). These workflows are specifically designed to process amplicon sequencing data generated according to the Midnight [19] or Artic protocols [9]. These protocols outline the wet-lab steps for PCR tiling of SARS-CoV-2 viral RNA from patient samples, followed by sequencing and basecalling using the Oxford Nanopore MinION sequencer. The resulting raw reads of basecalled ONT sequencing data are in FASTQ format and serve as input files for the two workflows.

#### 4.1.1. Workflow Structure

The bioinformatics analysis in both workflows consists of four main steps (Figure 3). Although each step has predetermined settings, these can be edited in Geneious Prime to customise the analysis.

In the first step, raw reads in FASTQ format undergo preprocessing using BBDuk (version 38.84) [20]. This preprocessing involves removing reads with an average Phred quality score lower than 10 or a total length shorter than 50 base pairs (bp). Primer sequences present among the target sequences for analysis are trimmed from both ends.

The second step maps the sequencing reads against a reference sequence using Minimap2 (version 2.17) [21]. The reference sequence we provide is the complete genome of SARS-CoV-2 Wuhan-Hu-1 (NCBI accession number: NC_045512.2).

In the third step, a built-in function in the Geneious software (version 2021.1.1) is used to generate a consensus sequence from the alignment. The consensus threshold is set to 50% by default, meaning that a base must be present in at least 50% of the aligned reads at a given position to be included in the consensus sequence.

In the fourth step, the consensus sequence undergoes annotation to identify the presence of genes and mutations of interest within the sample. The ‘Annotate from Database’ function within the Geneious software (version 2021.1.1) searches for matches between the consensus sequence and annotations stored in a user-specified database. Users may employ their own set of annotation features relevant to their analyses. We provide a dataset for testing the workflows containing common mutations diverging from the SARS-CoV-2 Wuhan-Hu-1 whole genome (Table 2). As new variants and functional mutations have arisen throughout the COVID-19 pandemic, relevant annotation features have been successively added to our dataset. Some examples of annotation features we provide include the following:Mutations distinguishing unique variants, such as Alpha, Beta, Delta, and Omicron.Functional mutations in the coronavirus spike glycoprotein.Mutations providing antiviral resistance against Ritonavir-boosted nirmatrelvir (Paxlovid) or Remdesivir in the coronavirus non-structural protein (nsp) 5 and 12.

These four main steps constitute the entire workflow ‘SARS-CoV-2_Assembly_Basic’, while ‘SARS-CoV-2_Assembly_WrapperPlugins’ includes additional steps that utilise two wrapper plugins (see Section 4.2. Plugins).

### 4.2. Plugins

We developed two wrapper plugins that encapsulate the command-line applications of the bioinformatics tools pangolin [14] and Nextclade [15], designed to provide phylogenetic assignment for the test samples. They are available as gplugin files and are manually imported into Geneious Prime. They were created to be implemented as the final steps in the ‘SARS-CoV-2_Assembly_WrapperPlugins’ workflow. However, once imported into Geneious, the plugins may also be directly used for analysing test data without being part of a workflow.

The pipeline of pangolin assigns the most likely SARS-CoV-2 lineage for the input sequences that pass the pipeline’s quality control check [14]. The resulting pango-lineage assignments, alongside quality control information and software version details, are stored in a CSV file as the output. Pangolin runs on macOS and Linux, and to ensure functionality on Windows, Docker is utilised within the plugin. The Nextclade pipeline contains various steps, such as sequence alignment and mutation calling, among others [15]. The combination of obtained signature mutations defines the assigned clade for a sequence. Clade assignments, along with other analysis results generated throughout Nextclade’s algorithm, are saved in a TSV/CSV file. In addition to the report of analysis results, the output includes files generated by the various software components in the pipeline (Output files—Nextclade documentation (nextstrain.org), accessed on 21 December 2023).

As wrapper plugins in Geneious Prime, pangolin and Nextclade generate the analysis results in tabular format which are subsequently imported back into the Geneious software. If the plugins are executed within the workflow, the input file is the generated consensus sequence. The analysis results are also exported to a user-selected folder on the computer. Similarly, the various output files from Nextclade are saved in a designated folder chosen by the user.

### 4.3. Set-Up of the Workflows

#### 4.3.1. Software

The software necessary for both workflows includes Geneious Prime, which executes the workflows and its functions, along with BBDuk for trimming and Minimap2 for sequence alignment. The latter two software are freely available as plugins on Geneious’s website, where they can be downloaded and imported into the software. The advanced workflow that also provides phylogenetic assignment (‘SARS-CoV-2_Assembly_WrapperPlugins’) requires additional software: pangolin (version 4.0 or later), Nextclade CLI (version 2.14.0 or later), and Python 3 (Table 3).

#### 4.3.2. Files

To run the basic main steps of the workflow, the required files are the primer scheme used for trimming, a reference genome for mapping the sequencing reads, and an annotation database. For SARS-CoV-2 analysis of ONT sequencing data, the associated files, listed in Table 2, are available on GitHub https://github.com/clinical-genomics-uppsala/Geneious_SARS-CoV-2 (accessed on 5 December 2023) and can be directly cloned and imported into Geneious Prime.

The two primer schemes provided correspond to the synthetic PCR primers used in the Artic and Midnight protocols. The primer schemes for the Artic and Midnight protocol, respectively, are available at https://github.com/artic-network/primer-schemes (accessed on 3 June 2024) and https://github.com/osilander/midnight-artic-1200 (accessed on 3 June 2024). The reference sequence utilised for alignment is the complete genome of the SARS-CoV-2 Wuhan-Hu-1, the NCBI Reference Sequence with accession number NC_045512.2. The set of annotation features includes mutations unique to specific variants and functional mutations observed in SARS-CoV-2.

Two separate GitHub repositories contain the plugins for pangolin (Geneious_pangolin_wrapper) and Nextclade (Geneious_Nextclade_wrapper). Users can download the plugins as gplugin files and install them directly into Geneious Prime. Additionally, we provide the source code for the plugins, giving users the option to build them in Geneious Prime from scratch using the ‘Wrapper Plugin Creator’ plugin.

The wrapper plugin for pangolin consists of a Python script that executes the Docker image for pangolin (covlineages/pangolin). This script checks for the latest image and updates the data every time the plugin is run. The use of Docker allows the same gplugin file for pangolin to run seamlessly across Linux, macOS, and Windows. The wrapper plugin for Nextclade is executed via Python. It downloads Nextclade’s latest dataset (https://docs.nextstrain.org/projects/nextclade/en/stable/user/datasets.html, accessed on 21 December 2023), which is necessary to run its pipeline. Additionally, the wrapper plugin creates a unique folder for Nextclade’s output files. Among these files are the analysis results in tabular format, containing Nextclade’s standard output information, as well as the versions of the Nextclade software and dataset used. For Windows, there is a unique Nextclade plugin as it contains the necessary Batch script to launch the Python script.

#### 4.3.3. Workflow Installation and Usage

After attaining the required software and files, the workflow(s) is ready to be set up in Geneious Prime. Detailed instructions for workflow installation and usage are available on GitHub, and as video tutorials on the research group’s YouTube channel (@EpiTaxEvo).

Among the files downloaded from the GitHub repository, the whole folder named ‘SARS-CoV-2_public_workflows’ is imported into Geneious, which places the files in the intended folder structure within the platform. If BBDuk and Minimap2 are not already present in Geneious, they can be installed by importing their Geneious plugins into the software. If pangolin and Nextclade are planned to be used, their gplugin files obtained from GitHub are also imported. The workflow files, with the file extension ‘.geneiousWorkflow’, are imported into Geneious as workflows.

Before the workflows can run, certain steps need to be edited. In the step ‘Trim using BBDuk’, the folder in Geneious with relevant primers is specified. In the step ‘Align/Assemble -> Map to Reference’, the correct reference sequence file, mapper software (Minimap2), and Data Type must be specified. In the step ‘Annotate from Database’, the folder in Geneious containing relevant annotation features is chosen. In the step(s) called ‘Export’, the user selects which folder on the computer to save the analysis results to.

If using the ‘SARS-CoV-2_Assembly_WrapperPlugins’ workflow, the steps executing the wrapper plugins for phylogenetic assignment need to be manually added to the workflow. Once added, the pangolin step requires specifying the path to the Docker executable on the computer. The Nextclade step needs the path to the Nextclade software and the path to desired output location on the computer specified.

To run the workflow, import the FASTQ files to be analysed into Geneious Prime. Select the files and start the workflow. Within the folder with test data on the Geneious software, the workflow creates subfolders that contain the alignments and consensus sequences. The consensus sequences are also saved to the computer as FASTA files. If pangolin and Nextclade are used, a third subfolder in the Geneious software is created to contain their analysis results. The same analysis results are also saved to the computer as txt-files in a designated folder. The Nextclade plugin creates a new folder on the computer to save Nextclade’s output files.

### 4.4. Validation and Test Data

The basic workflow, ‘SARS-CoV-2_Assembly_Basic’, underwent validation at Uppsala University Hospital in 2021 to ensure its suitability for both clinical application and national surveillance purposes. It adheres to the guidelines established by the European Centre for Disease Prevention and Control for SARS-CoV-2 sequencing and analysis [22].

The wrapper plugins were initially tested on Linux, macOS, and Windows, separately from the workflow, to validate their functionality on different operating systems and ensure the generation of the expected output. The complete genome of SARS-CoV-2 Wuhan-Hu-1 was analysed using the plugins in Geneious Prime, with the expectation of being assigned the pango-lineage B.

In February 2023, the enhanced workflow featuring the phylogenetic assignment, ‘SARS-CoV-2_Assembly_WrapperPlugins’, was compared to the basic workflow on a computer running Windows 10 Enterprise and Geneious Prime version 2021.1.1. The validation data consisted of the ONT data from 96 clinical samples, originally sequenced following the Midnight protocol, and analysed using the validated basic workflow. The pango-lineages had been obtained using the web applications of pangolin and Nextclade, serving as references for the output from the enhanced workflow. The certainty provided by pangolin and Nextclade has been established by comparing their results with lineages assigned by GISAID [23]. From August 2021, Uppsala University Hospital has uploaded 13,041 sequences obtained from the basic pipeline analysing clinical samples, whereas the 100% of the uploaded sequences have gotten the same lineage assigned by GISAID as the web interfaces of pangolin and Nextclade. To ensure that both presented pipelines are compared without any influence from sequencing or sample preparation variations, we used the same sequence dataset (FASTQ) for all analysis in this study. The 96 raw data sets obtained from a Nanopore sequencing run comprised 6.8 million sequence reads, resulting in 2.81 gigabases called, with a total size of 88.44 GB. An overview of how the workflows have been validated is presented in Figure 4.

The workflow execution times were measured. The run time of the basic ‘SARS-CoV-2_Assembly_Basic’ workflow included the analysis time taken for the workflow itself, along with the additional time required to obtain the pango-lineages of the consensus sequences using the web applications of pangolin and Nextclade. The time measured to run ‘SARS-CoV-2_Assembly_WrapperPlugins’ solely accounted for the time taken for the workflow to run.

For user convenience, we offer test data accessible from the SRA at NCBI (Accession number: PRJNA1048178). Its purpose is for users to control their installed workflow. This dataset consists of 18 samples of ONT sequencing of SARS-CoV-2 tiled amplicons obtained from former patient samples at Uppsala University Hospital. The selection covers a variety of SARS-CoV-2 lineages to ensure diversity. To secure the privacy rights of human participants, any human DNA sequences were removed before public release. The raw reads from the patient samples were aligned with the SARS-CoV-2 reference sequence (NC_045512.2) using Minimap2 in Geneious Prime. Unaligned reads were subsequently discarded. Successfully aligned reads were saved as FASTQ files. To ensure the removal of human DNA, the filtered reads were mapped against both the SARS-CoV-2 reference sequence and the human reference genome assembly GRCh38 using FastQ Screen (version 0.15.3) [24]. The filtered reads were then analysed with the ‘SARS-CoV-2_Assembly_WrapperPlugins’ workflow to verify that pangolin and Nextclade could correctly assign the same lineage to each sample as for the unfiltered reads.

## Figures and Tables

**Figure 1 ijms-25-06645-f001:**
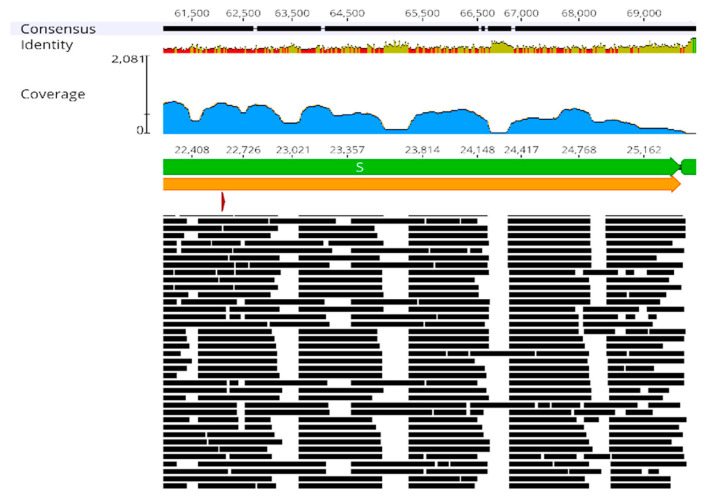
Alignment of Nanopore reads (black lines) from a single sample to the reference SARS-CoV-2 genome (NC_045512) following processing through the basic workflow. Geneious Prime displays the annotated genes (green) and open reading frames (orange), the read coverage (blue), and the consensus sequence (black, top part). The screenshot from Geneious Prime 2022.1.1 presents a partial view of the results.

**Figure 2 ijms-25-06645-f002:**
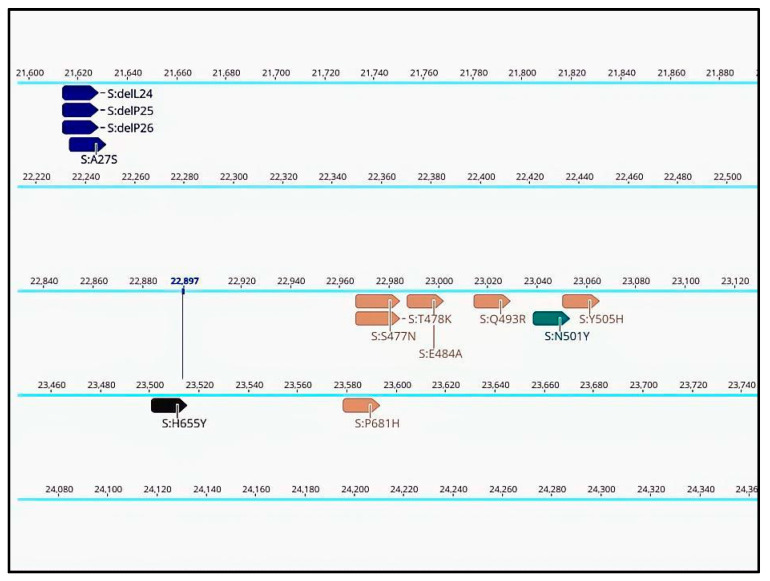
Annotated consensus sequence displayed in Geneious Prime 2022.1.1, derived from SARS-CoV-2 sequencing data processed through the basic workflow (partial view of the results).

**Figure 3 ijms-25-06645-f003:**
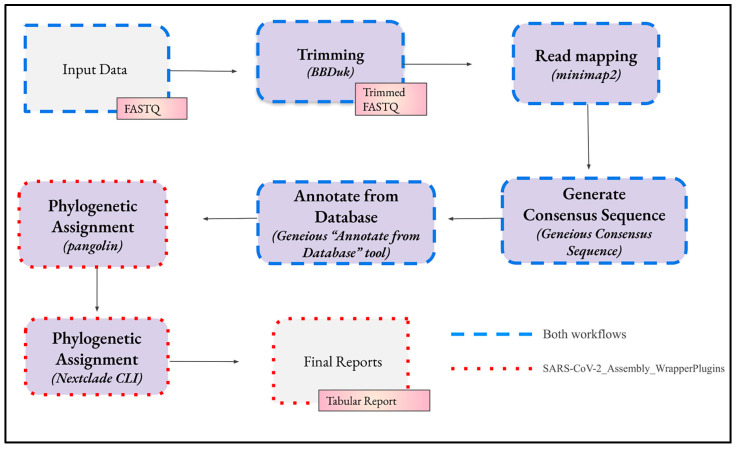
The sequential steps, depicted by the arrows, involved in the workflows for bioinformatics analysis of SARS-CoV-2 genomes in Geneious Prime. The imported FASTQ files undergo trimming, alignment to a reference sequence, transformation into a consensus sequence, and annotation to reveal potential mutations in the genomic sample. Phylogenetic assignment, performed by the tools pangolin and Nextclade, can be executed manually in their web interfaces or automatically by their CLI as part of the workflow.

**Figure 4 ijms-25-06645-f004:**
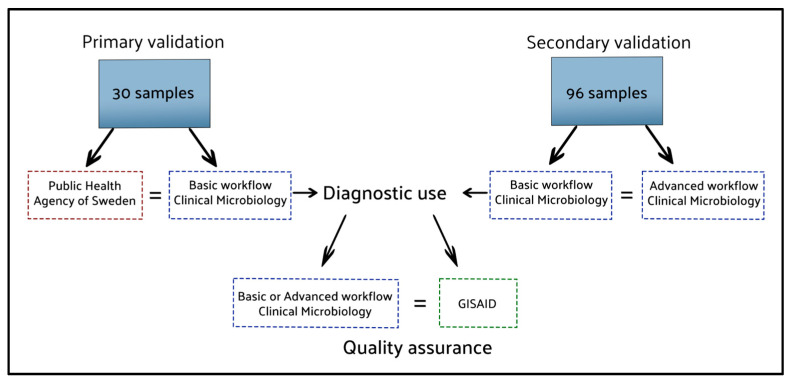
The results of the analysis methods compared to secure the workflows’ reliability. Blue dotted lines: workflows tested in our laboratory; Red and Green dotted lines: external references.

**Table 1 ijms-25-06645-t001:** The assigned lineages of a sub-selection of SARS-CoV-2 sequencing data after being analysed with the basic workflow + the web interfaces of pangolin and Nextclade or analysed with the advanced workflow where pango-lineage assignment is included as plugins.

Sample	Lineage (Pangolin Web Interface)	Lineage (Pangolin Plugin)	Lineage (Nextclade Web Interface)	Lineage (Nextclade Plugin)
1	B.1.1.7	B.1.1.7	B.1.1.7	B.1.1.7
4	AY.9.2	AY.9.2	AY.9.2	AY.9.2
5	AY.94	AY.94	AY.94	AY.94
6	AY.122	AY.122	AY.122	AY.122
7	BA.1	BA.1	BA.1	BA.1
9	BA.2	BA.2	BA.2	BA.2
10	BA.4.2	BA.4.2	BA.4.2	BA.4.2
11	BA.5	BA.5	BA.5.3	BA.5.3
13	BQ.1	BQ.1	BQ.1	BQ.1
14	BQ.1.1	BQ.1.1	BQ.1.1	BQ.1.1
15	XBB.1.5	XBB.1.5	XBB.1.5	XBB.1.5
16	XBB.1.9.2	XBB.1.9.2	XBB.1.9.2	XBB.1.9.2
17	CH.1.1.19	CH.1.1.19	CH.1.1.19	CH.1.1.19
18	EG.5.1.1	EG.5.1.1	EG.5.1.1	EG.5.1.1

**Table 2 ijms-25-06645-t002:** Required data files for setting up a functioning workflow in Geneious Prime. The files are available on a GitHub repository.

File	Purpose
SARS-CoV-2_Assembly_Basic.geneiousWorkflow	The basic workflow without pango-lineage assignment.
SARS-CoV-2_Assembly_WrapperPlugins.geneiousWorkflow	The workflow that includes pango-lineage assignment.
SARS-CoV-2 reference sequence.geneious (NC_045512)	SARS-CoV2 Wuhan-Hu-1 reference sequence for alignment including gene annotations.
Annotation features (Dec 2023 version).geneious	Example database of common mutations.
Primer SARS-CoV-2 Artic.geneious	Artic primers that are to be trimmed from the data.
Primer SARS-CoV-2 Midnight1200.geneious	Midnight primers that are to be trimmed from the data.

**Table 3 ijms-25-06645-t003:** The software requirements for either both or just the “SARS-CoV-2_Assembly_WrapperPlugins” workflow.

Software	Workflow	Link
Geneious Prime (version 2021.1.1 or later)	Both	geneious.com (accessed on 17 October 2023)
Python 3	“SARS-CoV-2_Assembly_WrapperPlugins”	python.org (accessed on 5 December 2023)
Docker Desktop (version 4.9.0 or later)	“SARS-CoV-2_Assembly_WrapperPlugins”	docker.com (accessed on 5 December 2023)
Nextclade CLI (version 2.14.0 or later)	“SARS-CoV-2_Assembly_WrapperPlugins”	nextstrain.org (accessed on 21 December 2023)

## Data Availability

Workflows available at https://github.com/clinical-genomics-uppsala/Geneious_SARS-CoV-2 (accessed on 5 May 2024). Sequence test data for workflow validation available at NCBI (Accession number PRJNA1048178). Instruction videos available at the research group’s YouTube channel (@EpiTaxEvo). Informed consent was obtained from all subjects involved in the study. All raw data processing was conducted in a clinical setting for method development. This study used only existing sequence data, with all consensus files having been made publicly available beforehand.

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
