# Peer review of "From SARS-CoV-2 to Global Preparedness: A Graphical Interface for Standardised High-Throughput Bioinformatics Analysis in Pandemic Scenarios and Surveillance of Drug Resistance"

_ijms, 2024, doi:10.3390/ijms25126645_

Round 1

Reviewer 1 Report

Comments and Suggestions for Authors

1-Number of samples were prepared and tested were not enough to study mutations and investigate behavior of SARS-CoV-2 with different variants alpha, beta, delta and omicron.  

2- In this study should be presented ethical approval of human samples.

3- In methods, the table of primers were used should be presented.

4- In results, table of mutations were investigated should be recorded with its pathogenic effect and sensitivity to the drugs.

5- The results should be presented in more than one figure to be more accurate and easy to understand.

Finally the title had Surveillance of Drug Resistance but the paper not had information about this point.

Reviewer 2 Report

Comments and Suggestions for Authors

Cumlin et al. describe a successful NGS workflow using a graphical user interface (GUI) in Genious Prime for analyzing and lineage typing Oxford Nanopore Technology SARS-CoV-2 data. The explosion in sequencing capabilities within the last two decades has forced experts in the fields of computer science and biology to integrate their disciplines to take advantage of the new technology. For end-users not familiar with command-line programming, GUI’s can be developed to augment the backend interaction between the user and program language. Workflows, such as the one described in this manuscript, can be easily adapted for genotyping and monitoring future pandemic microbes, assessing antibiotic- or drug-resistance genes, and automates repetitive tasks. The manuscript is well-written, highlights the real-world need for workflows such as that described, and provides understandable methods that enable the reader to perform similar analyses. The clinical utility of NGS, particularly related to infectious diseases, has been hampered by unclear validation standards. To improve the clarity and substantiate the validity of the workflow, consider addressing the below comments.

- The authors note the original basic analysis was validated against the method used by the Public Health Agency of Sweden. What parameters were established for validating the analysis?

- On a related note, the text states the advanced workflow was validated using 96 clinical samples, but there is no specification on what the validation parameters were.

- Utilizing NGS for infectious disease has primarily been used in surveillance and not diagnostics. The authors state validation resulted in accurate diagnoses that matched the results obtained from the validated basic workflow. What is meant when the authors state “diagnoses” and what is being compared between the advanced workflow validation and the original validated basic workflow?

- Under workflow validation, 96 clinical samples were used for validating the advanced workflow with plugins. Will the authors elaborate on what these clinical samples are? It is unclear if the 96 samples were re-extracted, library prepped, sequenced, and then underwent analysis using the workflows or if the 96 samples were simply raw FASTQ files. Do these samples represent a range of viral load, sample type, storage conditions, etc.?

- A visual summary of the validation results may help some readers better appreciate the work that has been done.
